

# Silurian syn- and post-collision granitic magmatism in the western section of the North Qinling Orogen: Implications for collisional orogenic processes

Hao Lin[1], Zuochen Li[1,2], Xianzhi Pei[1,2], Shaowei Zhao[1,2], Meng Wang[3], Hai Zhou[1,2], Feng Gao[4], Mao Wang[1], Li Qin[1]

[1]School of Earth Science and Resources, Chang'an University, Xi'an 710054, China
[2]Xi'an Key Laboratory for Mineralization and Efficient Utilization of Critical Metals, Xi'an 710054, China
[3]Geophysical Exploration Academy of China Metallurgical Geology Bureau, Baoding 071051, China
[4]School of Geology and Mining Engineering, Xinjiang University, Urumqi 830046, China

*Correspondence to*: Zuochen Li (lizuochen@chd.edu.cn)

**Abstract.** The Liqiao and Xianping plutons can provide crucial evidence for the collision-orogeny process of the Proto-Tethys Ocean in the western section of the North Qinling Orogen. In this study, we present petrological, zircon U-Pb geochronological, geochemical, and zircon Lu-Hf isotopic data for these plutons. Both the Liqiao and Xianping plutons are characterized as high-K, calc-alkaline, metaluminous to weakly peraluminous granites, with ages of 429 Ma and 421 Ma, respectively. The Liqiao pluton was classified as I-type granite, displaying positive $\varepsilon_{Hf}(t)$ values ranging from -0.1 to +3.4, and high Mg$^{\#}$ values from 37.86 to 48.25. We interpret this to indicate that it was generated by the partial melting of juvenile felsic lower crust, with a contribution from mantle-derived material. In contrast, the Xianping pluton exhibited lower Mg$^{\#}$ values (20.40 to 35.11) and negative $\varepsilon_{Hf}(t)$ values (-18.0 to -13.9), consistent with the geochemical characteristics of highly fractionated I-type granite. This suggests that the Xianping pluton formed through the partial melting and extensive fractional crystallization of ancient felsic crust. We propose that the Liqiao pluton originated in a syn-collisional setting, while the Xianping pluton formed in a post-collisional environment. Both plutons are products of the collisional orogeny between the Yangtze Block and the North Qinling Orogen, which were associated with the closure of the Wushan-Shangdan Ocean, the northern of the Proto-Tethys Ocean.

## 1 Introduction

It is generally believed that the Proto-Tethys Ocean was a giant ocean, which was located between the northern Laurasia continent and the southern Gondwana continent (Stampfli and Borel, 2002; Li et al., 2016a; Wu et al., 2020). The Central China Orogenic Belt (CCOB) is one of the most developed areas of the Proto-Tethys tectonic domain, it includes the West Kunlun Orogen, East Kunlun Orogen, Altyn Orogen, North Qaidam Orogen, Qilian Orogen and Qinling-Dabie Orogen, which preserved abundant information on the process of orogeny of various blocks in the North China, South China, Qaidam,



Tarim, and Qiangtang regions from the Paleozoic-Early Mesozoic (Dong et al., 2021, 2022a, 2022b). The Qinling Orogen (QO) is an important component of the CCOB, which was located between the Yangtze Block (YB) and the North China Block (NCB), it consists of two sutures (the Wushan-Shangdan Suture (WSS) and the Mianlue Suture (MS)) and three blocks (the North Qinling Orogen (NQO), the South Qinling micro-block, and the northern margin of the YB) (Zhang et al.,
2001; Wang et al., 2013, 2015; Dong et al., 2021, 2022a, 2022b; Fig. 1a). Previous scholars have conducted extensive studies on the QO (Zhang et al., 2001, 2019; Pei et al., 2004, 2009; Wu et al., 2006; Wang et al., 2006, 2007; Xu et al., 2008; Dong et al., 2011a, 2011b, 2011c, 2021), the Early Paleozoic was the main ocean-continent transformation stage of the QO (Xia et al., 1998; Dong et al., 2008; Wang et al., 2015; Ren et al., 2019; Dong et al., 2021). As an important part of the QO, the NQO has complex tectonic evolution and large-scale magmatism. It is a key for exploring the collision of the YB and the
NQO. Also, it is an important window for studying the structural evolution of the Proto-Tethys Ocean.

Existing studies indicate that the WSS was formed during the Caledonian period., which is the result of the collision orogeny between the YB and the NCB (Ren et al., 2019). However, there is still considerable controversy over the duration of the subduction-collision orogeny process of the northern branch of the Proto-Tethys Ocean represented by the Wushan-Shangdan Ocean (WSO). There are two main views: some scholars believe that the main orogenic stage of the NQO was the
late Neoproterozoic to the Mid-Late Triassic, the WSO was subducted into the NCB during the Early Paleozoic, by the Middle-Late Devonian, this process transitioned into a continent-continent collision between the YB with the NCB, culminating in the closure along the WSS (Zhang et al., 2001, 2019; Dong et al., 2011a, 2011c). Another scholar believes that the WSO was formed in the NQO during the Late Cambrian, the ocean subduction and the development of ancient island arc occurred in the Ordovician, until to the Silurian-Late Devonian periods, there were continent-continent or
continent-arc collision orogeny (Pei et al., 2009; Wang et al., 2009; Ren et al., 2019).

In the western section of the NQO, there are lots of Caledonian basic-intermediate pluton and granite distributed in the Liushuigou-Shuangchangxia and Baihua-Liqiao areas (Pei et al., 2007a; Fig. 1b). These are primarily magmatic rocks related to subduction-collision (Dong et al., 2011a; Li et al., 2018a; Ren et al., 2018, 2021; Yang et al., 2018a). Research on the Paleozoic magmatic rocks of different ages and origins in the western section of the NQO can provide important evidence
for the subduction-collision-post-collision orogeny process of the WSO in the western section of the NQO in the Early Paleozoic, there by constraining the process and timing of Early Paleozoic orogeny in eastern of the Proto-Tethys Ocean. For a clearer understanding of the tectonic evolution of the western section of NQO, we focus on the Liqiao pluton (LP) and Xianping pluton (XP) in the western section of the NQO (Fig. 1, 2). We conduct studies on petrology, geochemistry, zircon U-Pb geochronology, and zircon Lu-Hf isotope to determine the petrogenesis and age of the rocks, trace the magma source
and tectonic background, establish a sequence of tectonic and magmatic evolution events, and explore the relationship between the Caledonian orogeny in the NQO and the tectonic evolution of the Proto-Tethys Ocean.





**Figure 1: (a) Simplified geological map of the division of tectonic units in Qinling Orogen (Dong et al. 2022) (b) Simplified geological map of the conjunction zone between the NQO and NQLO showing the distribution of Early Paleozoic plutons (modified from 1:500,000 geological map of Qinling metallogenic belt, and Bottom image from Map World)**



## 2 Geological background

### 2.1 Regional geological background

The western section of the NQO is located in the central and western part of China, and is positioned at the conjunction between the NQO and the North Qilian Orogen (NQLO) in the middle section of the CCOB (Zhang et al., 2001; Dong et al., 2022a; Fig. 1a). It is limited in the Tethys tectonic domain, the Paleo-Asian Ocean tectonic domain, and the Pacific tectonic domain, which has developed an active continental margin trench-arc-basin system in the Early Paleozoic (Pei et al., 2009; Dong et al., 2011a; Zhang et al., 2011; Dong and Santosh, 2016). The ophiolite with Late Cambrian N-MORB type basic

volcanic rocks developed in the Guanzizhen area, which extends eastward to the Liqiao area and westward to the Wushan-Yuanyangzhen area, representing the material record of ancient oceanic crust (Pei et al., 2004, 2007a; Dong et al., 2008). In the Late Cambrian-Early Ordovician, the WSO (represented by the Guanzizhen-Wushan ophiolite) was subducted from south to north, and formed the island arc-fore-arc basin represented by the Liziyuan Group metamorphic sedimentary-volcanic rocks (Pei et al., 2006; Yang et al., 2018b). With the continued subduction of the WSO, the metavolcanic rocks of

the Caotangou Group and the corresponding volcaniclastic and shallowly metamorphosed clastic rocks, which are typical of island arc, were formed in the Middle-Late Ordovician (Pei et al., 2009). Concurrently, intermediate-basic igneous complex of Liushuigou and Baihua were formed (Pei et al., 2007b; Gao et al., 2012), as well as subduction-type pluton such as Tangzang quartz diorite, Honghuapu tonalite, Yangjiazhuang quartz diorite, and Sanchahe quartz diorite (Chen et al., 2002, 2008; Wang et al., 2006; Ren et al., 2018; Qin et al., 2022). The tectonic background began to transition to the continental-

continental or arc-continental collision orogeny after the subduction of the ancient oceanic crust and developed Caledonian collision-type pluton such as Dangchuan granite, and entered into an extension environment until the end of the orogeny in the Late Silurian-Early Devonian (Wang et al., 2008; Wang, 2013; Ren et al., 2018, 2021; Qin et al., 2022; Xin and Huang, 2023).

### 2.2 Regional tectonic and stratigraphic features

The western section of the NQO is bounded by the NQLO and NCB to the north and the South Qinling tectonic belt to the south (Fig. 1b). The exposed strata in this area range from the Paleoproterozoic to the Early Paleozoic (Pei et al., 2009). The Precambrian crystalline basement in the western section of the NQO are composed by Paleoproterozoic Qinling Group, which was composed by felsic gneisses, aluminous gneiss and marble-calc-silicate (Pei et al., 2009; Diwu et al., 2014). Previous zircon U-Pb dating indicate that it was formed in 2298 to 1867 Ma (Zhang et al., 2001). The high-pressure and

ultra-high-pressure (HP-UHP) metamorphic rock are exposed in the Qinling Group, which are the products of exhumation of continental crust experienced subduction-deep subduction during the Early Paleozoic (Gong et al., 2016; Liu et al., 2020). The Kuanping, Liziyuan and Caotangou Group (Fig. 1b) are consisted by metamorphic volcanic-sediment sequence. The previous studies of the Kuanping Group have mainly focused on the central and eastern parts of the NQOB, and have reported Meso-Neoproterozoic ages from 1974 to 813 Ma for the meta-volcanic rocks which with normal-type mid-ocean



ridge basalt (N-MORB) affinities, and it was considered to represent oceanic crust (Xue et al.,, 1996; Zhang et al., 2001; He et al., 2007a; Pei et al., 2009; Diwu et al., 2010; Gao et al., 2015; Dong et al., 2014, 2015, 2021; Zeng et al., 2023). The Paleozoic Liziyuan Group are composed by metamorphic clastic rocks and carbonates sedimentary facies and metamorphic basalt, metamorphic basalt andesites, and metamorphic andesites volcanic facies, while the volcanic displayed typical island-arc or fore-arc affinity (Pei et al., 2006). The Ordovician Caotangou Group was composed by metamorphic volcanic-sediment, and it was divided into the lower Honghuapu Formation, the middle Zhangjiazhuang Formation, and the upper Longwanggou Formation (Song et al., 1991; Sun and Dong, 1995; Pei et al., 2009; Chen et al., 2019), while the volcanic rocks showed island-arc affinity (Yan et al., 2007; Zhu et al., 2008; Xu et al., 2014; Xie et al., 2020). These Paleozoic volcanic-sediment sequence in the western section of the NQO indicated that the Early Paleozoic tectonic evolution of the NQO (Xu et al., 2008; Pei et al., 2009; Dong et al., 2015, 2021).

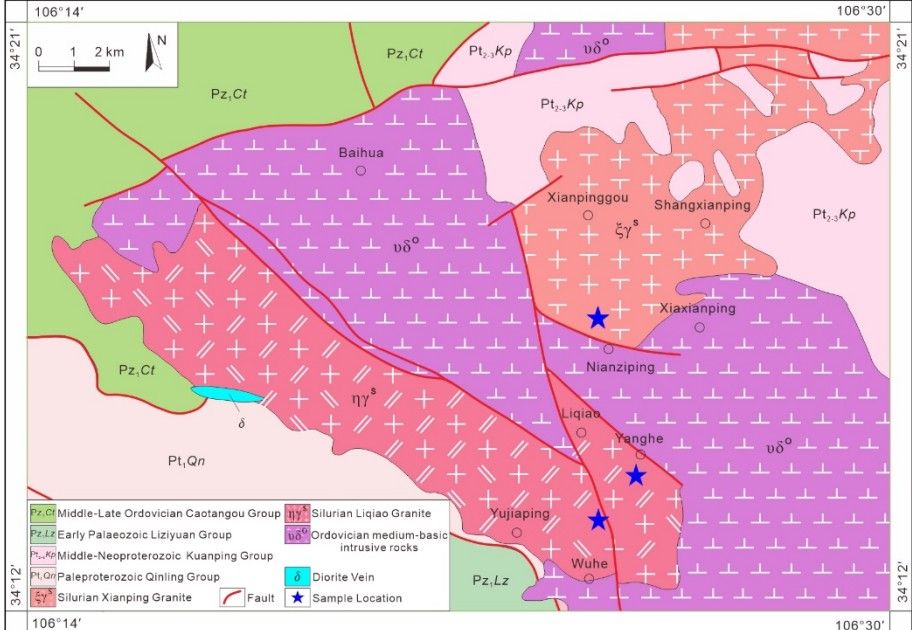

**Figure 2: Geological map showing the LP and XP in the western section of NQO**

## 3 Field geology and petrography

### 3.1 Liqiao pluton

The LP is located in the Xiongshangou-Shaping-Yujiaping-Liqiao area of the western section of the NQO (Fig. 1b, Fig. 2), and the pluton is distributed in the NWW direction, which is consistent with the regional tectonic line. In spatial distribution,



it intruded into the Paleoproterozoic complex (e.g. Qinling Group), the Early Silurian Baihua intermediate-basic igneous complex, the Ordovician metasedimentary clastic strata (e.g. the Honghuapu Formation of the Caotangou Group). It was

consisted by monzogranite and a few syenogranite (Fig. 3a-b). The red monzogranite shows medium- to coarse-grained, some with pseudoporphyritic granite, and massive structure (Fig. 3a). Samples from the monzogranite are composed by K-feldspar (35~45 vol%), plagioclase (35~45 vol%), quartz (20~25 vol%), biotite (~5 vol%). The accessory minerals are mainly apatite, sphene, ilmenite, monazite. It shows distinct myrmekitic structure (Fig. 3c). K-feldspar is euhedral, with development of gridiron twinning, ranging from 1.0 to 5.0 mm in size. Plagioclase is generally euhedral or subhedral, with

development of polysynthetic twin and carlsbad—al bite compound twin, ranging from 2.0 to 3.0 mm in size. The plagioclase commonly has crystallization of subhedral columnar. Quartz is allotriomorphic granular structure with undulatory extinction. Biotite is yellow-brown, block-like structure with significant pleochroism, local chloritization. To contrast with the monzogranite, the syenogranite (Fig. 3b) has higher K-feldspar (45~55 vol%), lower plagioclase (20~25 vol%), similar quartz (20~30 vol%), biotite (~5 vol%).

## 130   3.2 Xianping pluton

The XP locates in the east-nouth of the LP (Fig. 2). The pluton intruded into the Meso-Neoproterozoic metamorphic volcanic-sediment (e.g. Kuanping Group), the Early Silurian Baihua intermediate-basic igneous complex. It is bounded by the Qinlingdabao granite to the north, which is in contact with a fault. The pluton is mainly syenogranite, which shows medium-coarse-grained and massive structure (Fig. 3e-f). There was a distinct coarse-medium-grained transition zone can be

seen in the pluton (Fig. 3e). The syenogranites are comprising K-feldspar (45~60 vol%), plagioclase (10~20 vol%), quartz (25~30 vol%), biotite (~5 vol%), and minor accessory minerals include zircon, apatite, sphene, monazite, hornblende (Fig. 3g-h). K-feldspar is ranging from 2.0 to 4.0 mm. Plagioclase is ranging from 2.0 to 5.0 mm. Quartz is subhedral and allotriomorphic granular structure with undulatory extinction, it fills in the spaces between other minerals in a granular form. Biotite is gray-brown, block-like structure, form with significant pleochroism, local chloritization.

## 140   4 Sample collected and analytical methods

### 4.1 Sample collected

Two zircon U-Pb dating samples were collected from the LP, including a monzogranite (TS19001-3) and a syenogranite (TS19007-2). The locations of the two samples are 34°14'04.05"N, 106°25'09.71"E, and 34°13'22.92"N, 106°24'27.68"E in the Liqiao-Yanghe-Wuhe area. Two zircon U-Pb dating samples were collected from the XP, including two syenogranite

(TS19014-1 and TS19014-3). The sampling location is in the Xianpinggou-Shangxianping-Nianziping area, which located in 34°16′28.15″N, 106°24′26.88″E, and 34°16′28.21″N, 106°24′26.95″E, respectively. We also collected 15 whole-rock geochemical samples from the LP and 12 samples from the XP. All samples were fresh and free of corrosion, and the sampling locations are shown in Fig. 2.





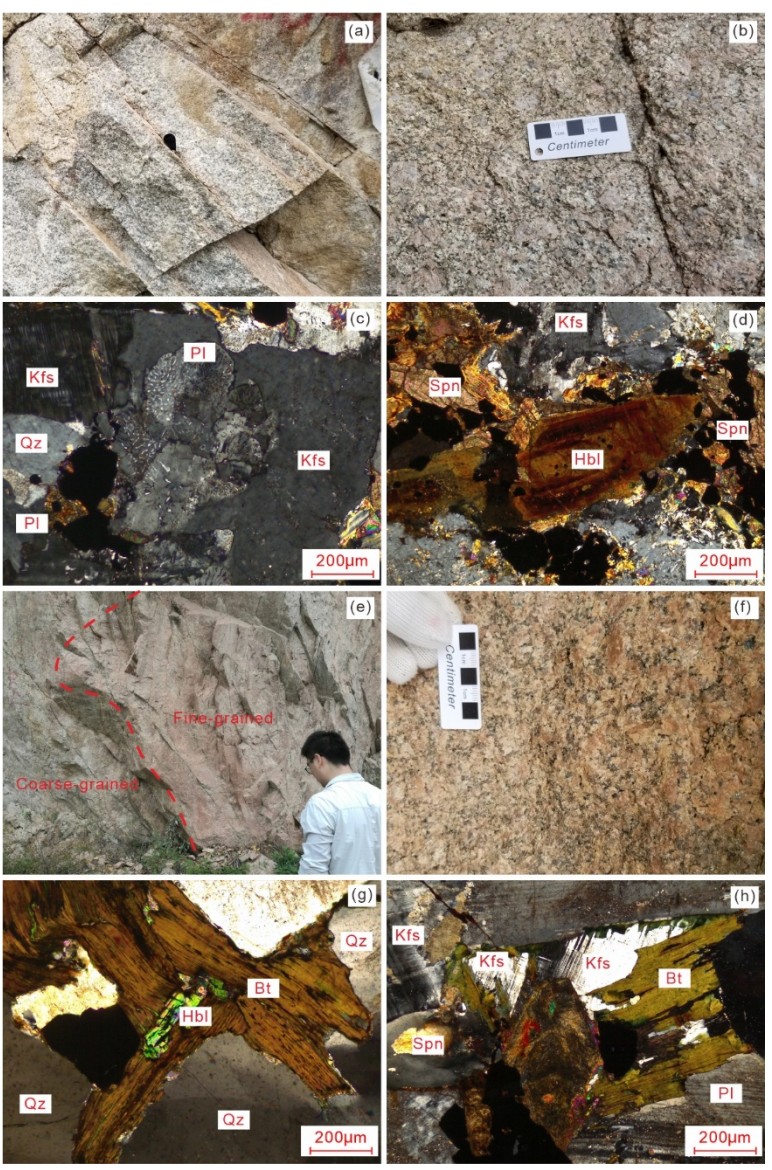

**Figure 3: Representative field photographs and photomicrographs of the LP and XP in the western section of NQO. Kfs = K-feldspar, Pl = Plagioclase, Bt = biotite, Qz = Quartz, Hbl = Hornblende, Spn = Sphene.**

## 4.2 Zircon U-Pb age

The samples used for geochronological research were crushed and zircon separated by Xi'an Ruishi Geological Technology
Co., Ltd. Zirconium target and cathodoluminescence (CL) imaging were completed by Beijing Zircon Nian Linghang Technology Co., Ltd. Zircon U-Pb isotope testing was conducted on the German Jena PQMS ICP-MS instrument using the laser ablation system NWR193. The 91500 standard sample and GJ-1 control sample were used in this analysis. Data




processing was done using the ICPMSDataCal program (Liu et al., 2010). Weighted mean age calculations and concordia diagrams were produced using the Isoplot program (Ludwig, 2012). For a detailed description of the analysis methods and

instrument parameters, refer to Li et al., (2009).

### 4.3 Whole-rock major and trace elements

The testing of major, rare earth, and trace elements in whole-rock samples was conducted in the Key Laboratory of Western China's Mineral Resources and Geological Engineering, Ministry of Education, Chang'an University. The analysis of major elements was performed using X-ray fluorescence spectroscopy (XRF) method. The XRF fusion method was carried out in

accordance with the national standard GB/T 14506.28-1993, with an analysis precision better than 2% to 3%. The samples were weighed after being heated at 1000 °C for 90 minutes in an oven to determine the loss on ignition (LOI). Rare earth and trace elements were analyzed using a Thermo-X7 Inductively Coupled Plasma Mass Spectrometer (ICP-MS).

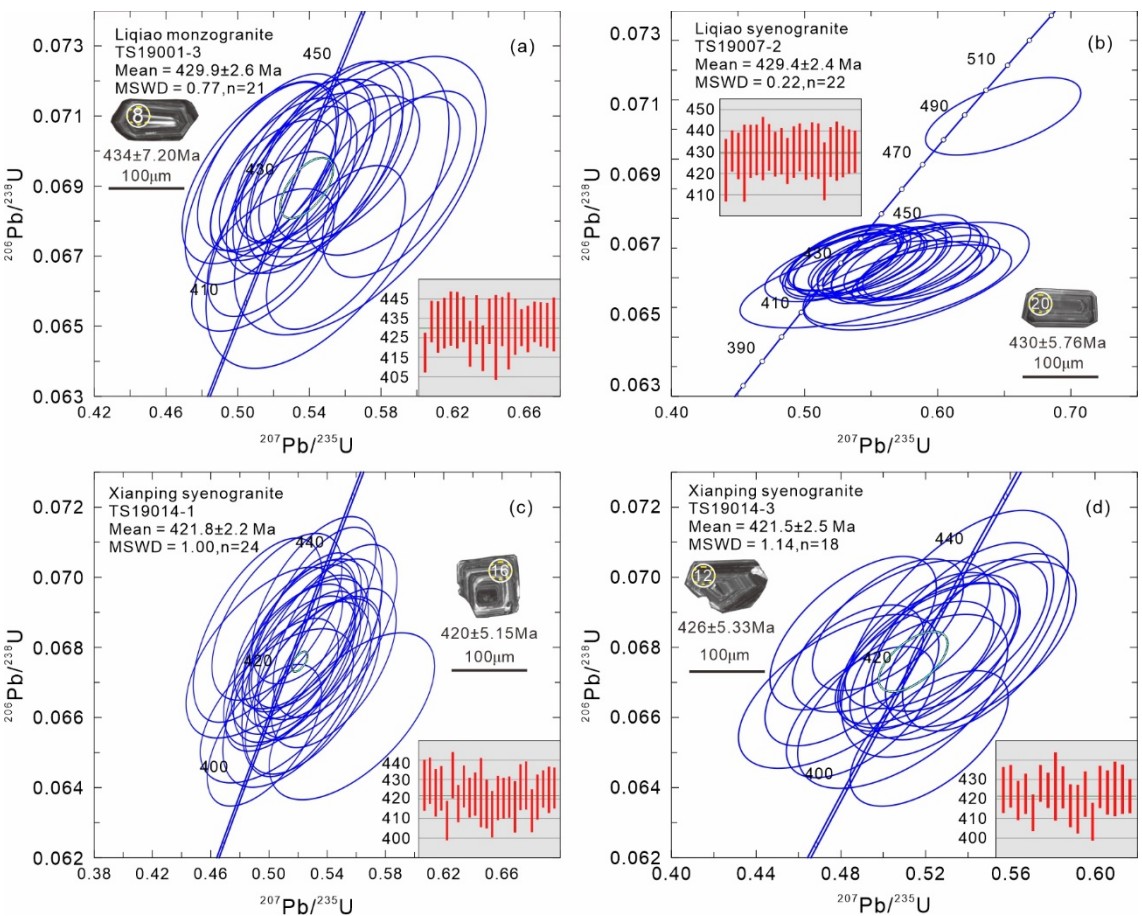

**Figure 4: CL image of zircons and the U-Pb zircon concordia diagrams from the LP and XP in the western section of NQO.**



## 4.4 Zircon Lu-Hf isotope

The zircon Lu-Hf isotope dating was carried out at Langfang Fengzeyuan Rock Mine Detection Technology Co., Ltd. The selected Zircon Lu-Hf isotope analysis point was located in the in-situ region of the zircon U-Pb dating site, and the zircon

was ablated by the Resolution SE 193 nm excimer laser ablation system of ASI (Applied Spectra Inc.), and the spot beam diameter of laser ablation was generally 38 μm, the energy density was 7~8 J/cm², and the frequency was 10 Hz. The analytical system used is the multi-collector inductively coupled plasma mass spectrometer (Neptune Plus) from the American company Thermo Fisher, with laser ablation material introduced into the Neptune Plus (MC-ICPMS) using high-purity He as carrier gas. The temperature requirement for the detection environment is 18~22℃, with a relative humidity of

less than 65%. For detailed experimental principles, analytical techniques, and experimental procedures, refer to Wu et al., (2007) and Geng et al., (2011).

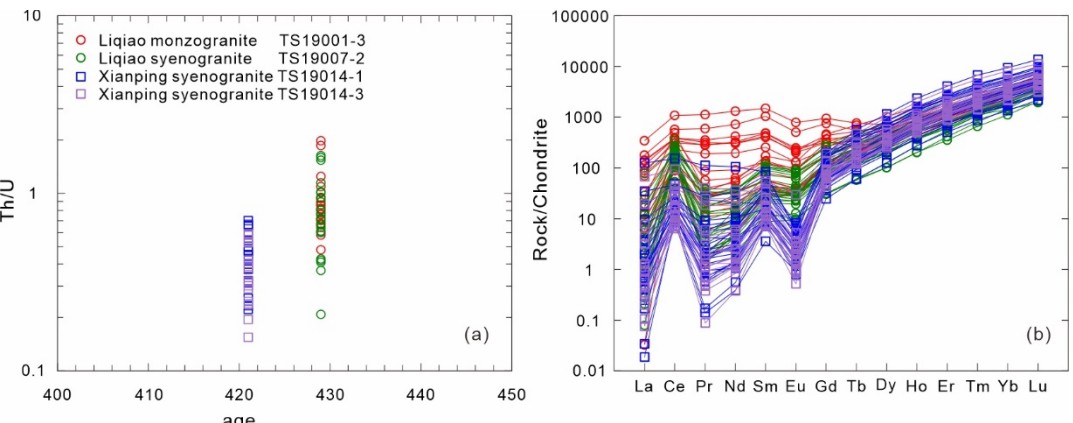

**Figure 5: Zircons U-Pb age-Th/U (a) and Chondrite-normalized REE patterns (b) diagram of the LP and XP in the western**
**section of NQO. The chondrite and Primitive Mantle values are from Sun and McDonough (1989).**

## 5 Results of analyses

The data for zircon U-Pb ages, zircon trace elements, whole-rock major and trace elements, and zircon Lu-Hf isotopes are shown in Supplementary Tables 1, 2, 3 and 4, respectively.

## 5.1 Zircon U-Pb age

The zircon CL image and concordant U-Pb diagrams are shown in Fig. 4. Zircons (TS19001-3) from the Liqiao monzogranite have length of 80 to 180 μm, with respect ratios from 1:1 to 3:1, most grains display developed oscillatory zoning (Fig. 4a). Except for 4 discordant spots, the remaining 21 concordant spots have U = 1020 to 2806 μg/g, Th = 875 to 4928 μg/g, with variable Th/U ratios of 0.48 to 1.98, consistent with the characteristics of typical magmatic zircons (Fig. 5).



They have $^{206}Pb/^{238}U$ ages of 417 to 435 Ma (Fig. 4a) and yielding a weighted mean age of 429.9 ± 2.6 Ma (MSWD = 0.77,
n = 21).

        Zircons (TS19007-2) from the Liqiao syenogranite have length of 60 to 170 μm, with respect ratios of 1:1 to 2:1, most
grains display developed oscillatory zoning (Fig. 4b). 23 out of 25 spots have U = 341 to 2939 μg/g, Th = 290 to 3153 μg/g,
with Th/U ratios of 0.21 to 1.63, consistent with the characteristics of typical magmatic zircons (Fig. 5). One spot (#21) has
$^{206}Pb/^{238}U$ age of 489±6.39Ma, suggesting inherited origin. The remaining 22 spots have $^{206}Pb/^{238}U$ ages of 421 to 432 Ma
(Fig. 4b) and yield a weighted mean age of 429.4 ± 2.4 Ma (MSWD = 0.22, n = 22).

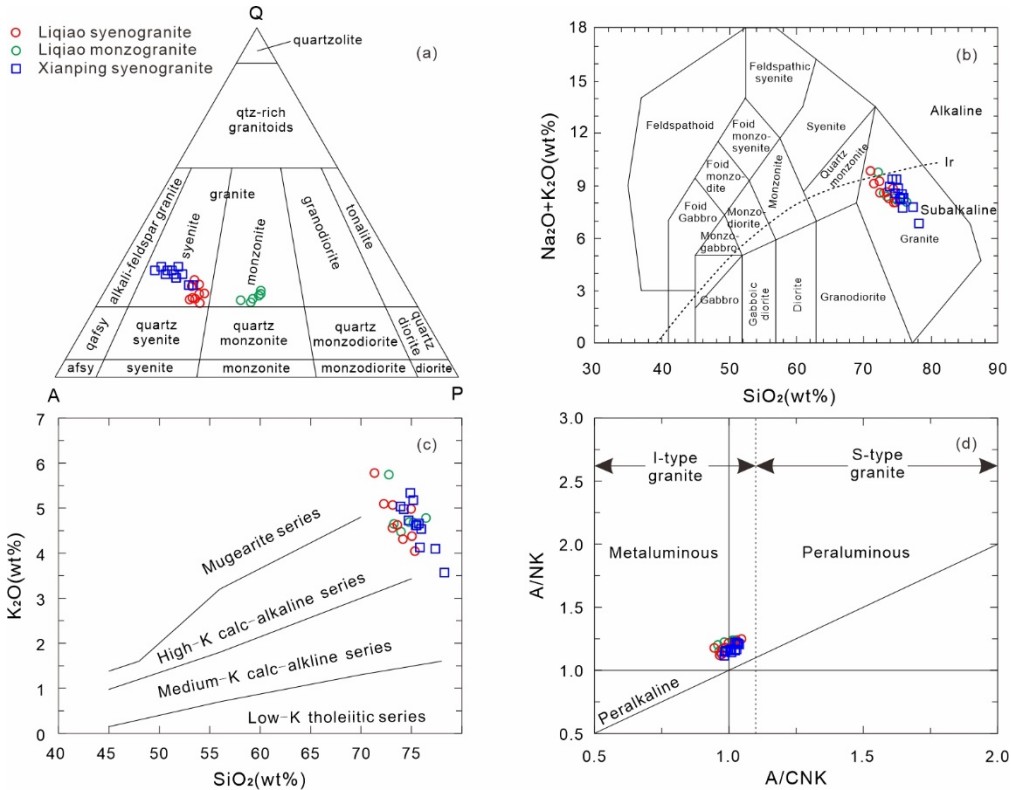

**Figure 6: Geochemical plots for rocks of the LP and XP of the western section of NQO: (a) Q-A-P diagram (Streckeisen 1976); (b)**
**TAS classification diagram (Peccerillo and Taylor 1976); (c) K₂O versus SiO₂ diagram (Middlemost 1994); (d) A/CNK (molar**
**Al₂O₃/[CaO+Na₂O+K₂O]) versus A/NK (molar Al₂O₃/[Na₂O+K₂O]) diagram (Maniar and Piccoli 1989).**

        Zircons (TS19014-1) from the Xianping syenogranite have length of 50 to 130 μm, with respect ratios of 1:1 to 3:1,
most grains display developed oscillatory zoning (Fig. 4b). Except for one disconcordant spots, the remaining 24 spot have
high U and Th contents (U = 692 to 9684 μg/g, Th = 342 to 3117 μg/g, with variable Th/U ratios of 0.22 to 0.70). They are
consistent with the characteristics of typical magmatic zircons (Fig. 5). These 24 spots have $^{206}Pb/^{238}U$ ages of 409 to 432 Ma
(Fig. 4b), yield a weighted mean age of 421.8±2.2 Ma (MSWD=1.00, n=24).



Zircons (TS19014-3) from the Xianping syenogranite have length of 40 to 170 μm, with respect ratios of 1:1 to 4:1, most grains display developed oscillatory zoning (Fig. 4c). In the 25 analysis spots, 18 spots display concordant U-Pb ages. They have high U and Th contents (U = 1244 to 12052 μg/g, Th =714 to 2351 μg/g, with variable Th/U ratios of 0.15 to 0.59), consistent with the characteristics of typical magmatic zircons (Fig. 5). These 18 spots have $^{206}Pb/^{238}U$ ages of 408 to 428 Ma (Fig. 4d), yield a weighted mean age of 421.5 ± 2.5 Ma (MSWD = 1.04, n = 18).

## 5.2 Whole-rock major and trace element geochemistry

### 5.2.1 Liqiao pluton

Liqiao monzogranite and syenogranite have similar geochemistry characteristics. The LP display high $SiO_2$ (71.09 to 76.45 wt%) contents, $Al_2O_3$ = 12.15 to 15.26wt%, as shown in the Q-A-P diagram (Fig. 6a), these sample plotted in the field of monzogranite and syenogranite. They have low $Fe_2O_3^T$ (0.90 to 1.77 wt%) contents, MgO (0.30 to 0.55 wt%) contents and $Mg^{\#}$ (37.86 to 48.25) values, high differentiation index (DI= 89.21 to 93.24) values, high $K_2O$ (4.02 to 5.76 wt%) contents, high total alkali($K_2O+Na_2O$= 8.02 to 9.83 wt%) contents and low $Na_2O/K_2O$(0.68 to 1.00) ratios. In the TAS diagram (Fig. 6b), they plotted in the subalkaline granite field. They have low rittmann index(σ) values of 1.94 to 3.43. As shown in the $SiO_2$ versus $K_2O$ diagram (Fig. 6c), most samples display high-K calc-alkaline series. All the samples have moderate A/CNK values of 0.95 to 1.05 (Fig. 6d).

All samples have high total REE contents of 180.35 to 412.32 ppm. They show variable enrichment in LREE with $(La/Yb)_N$ ratios = 27.46 to 51.51, $(La/Sm)_N$ ratios = 6.94 to 12.72, $(Gd/Yb)_N$ ratios = 2.49 to 3.48, and display insignificant Eu anomalies with Eu/Eu* = 0.54 to 1.03, Ce/Ce* = 0.90 to 0.98 (Fig. 7a). On the primitive mantle normalized trace element spider diagram (Fig. 7b), all these samples show enrichment in Rb, Th, K and Pb, and significantly negative Nb, Ta, Zr, Sr, P, Ce and Ti anomalies, with high Sr (229 to 336 ppm), Ba (688 to 14451 ppm) and Rb (202 to 267 ppm) contents, characterizing crust-derived melts.

### 5.2.2 Xianping pluton

Xianping syenogranite have $SiO_2$ = 71.09 to 76.45 wt%, $Al_2O_3$ = 12.15 to 15.26wt%, $Fe_2O_3^T$ = 0.56 to 1.91 wt%, MgO= 0.13 to 0.26 wt% and $Mg^{\#}$ = 20.40 to 35.11, differentiation index (DI) = 91.69 to 94.47. Samples from the XP plot into the syenogranite field on the Q-A-P diagram (Fig. 6a). In the TAS diagram (Fig. 6b), all the samples plotted in the subalkaline granite field. They are rich in $K_2O$ ($K_2O$= 3.57 to 5.34 wt%, $Na_2O/K_2O$ = 0.71 to 0.91) and have total alkali ($K_2O + Na_2O$) contents of 6.83 to 9.37 wt%. They have low rittmann index(σ) values of 1.32 to 2.81. The Xianping syenogranite show calc-alkaline affinity. As shown in the $SiO_2$ versus $K_2O$ diagram (Fig. 6c), most samples display high-K calc-alkaline series. Xianping syenogranite are metaluminous-weakly peraluminous with A/CNK ratios of 0.98 to 1.04 (Fig. 6d).



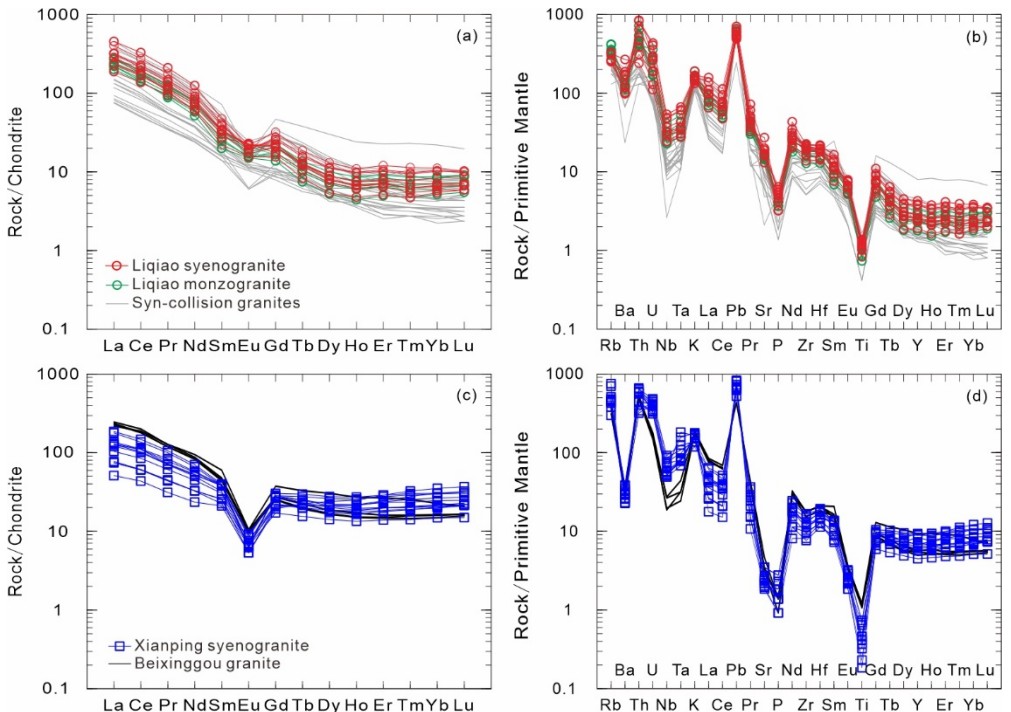

**Figure 7: Chondrite-normalized REE patterns (a, c) and primitive mantle normalized trace element spider diagrams (b, d) of the LP and XP in the western section of NQO. The chondrite and Primitive Mantle values are from Sun and McDonough (1989). Syn-Collision plutons from the western section of NQO after Wang et al. 2008; Ren et al. 2021; Qin et al. 2022; Xin and Huang 2023. Beixinggou pluton from the western section of NQO after Ren et al. 2021.**

These samples have REE contents of 81.26 to 205.05 ppm and show fractionated REE patterns expressed by LREE enrichment and HREE depletion with $(La/Yb)_N$ = 1.46 to 8.99, $(La/Sm)_N$ = 2.43 to 4.66, $(Gd/Yb)_N$ = 0.55 to 1.39. They show significant negative Eu anomalies with Eu/Eu* = 0.18 to 0.33 (Fig. 7c). On the spider diagram (Fig. 7d), they are enriched in Rb, K, Pb, Th, and U and depleted in Nb, Ta, Ti, P, Zr, Ce and Ba, with low Sr (39 to 74 ppm), Ba (159 to 271 ppm) and Rb (189 to 477 ppm) contents.

**5.3 Zircon Lu-Hf isotope**

Fifteen Lu-Hf spots from the Liqiao monzogranite (TS19001-3, 429 Ma) have positive $\varepsilon_{Hf}(t)$ values of +0.5 to +3.4, corresponding two-stage model ages of 1197 to 1383 Ma. Sixteen zircon Lu-Hf analysis spots from the Liqiao syenogranite (TS19007-2, 429 Ma) have variable Lu-Hf isotopic compositions, 14 spots display positive $\varepsilon_{Hf}(t)$ values of +0.6 to +3.3, corresponding two-stage model ages of 1202 to 1374 Ma, only one spot (#1) has negative $\varepsilon_{Hf}(t)$ value of -0.1 with corresponding two-stage model ages of 1411 Ma. And one inherited origin spot (#21) has positive $\varepsilon_{Hf}(t)$ values of +0.3, corresponding two-stage model ages of 1269 Ma.





Fifteen Lu-Hf spots from the Xianping syenogranite (TS19014-1, 421 Ma) have negative $\varepsilon_{Hf}(t)$ values of -18.0 to -13.9, corresponding two-stage model ages of 2276 to 2530 Ma. Fifteen Lu-Hf spots from the Xianping syenogranite (TS19014-3,

421 Ma) have negative $\varepsilon_{Hf}(t)$ values of -18.5 to -13.6 with corresponding two-stage model ages of 2259 to 2560 Ma.

**Figure 8: Pluton type discrimination diagram of the LP and XP in the western section of NQO: (a) FeO^T/MgO versus (Zr+Nb+Ce+Y) (Whalen et al. 1987); (b) (K_2O+Na_2O)/CaO versus (Zr+Nb+Ce+Y) (Whalen et al. 1987); (c)**
**100×(MgO+FeO^T+TiO_2)/SiO_2 versus (Al_2O_3+CaO)/(FeO^T+Na_2O+K_2O) (Sylvester 1989); (d) 10000Ga/Al versus Zn (Whalen et al. 1987); (e) 10000Ga/Al versus Zr (Whalen et al. 1987); (f) Zr versus TiO_2.**



## 6 Discussion

### 6.1 Types of rock genesis

Currently, the most widely used classification scheme for granite types is the MISA classification, with M-type granite
derived from mantle magmas being rare, and I-type, S-type, and A-type granite being predominant (Chappell and White, 1974, 2001; Coleman and Peterman, 1975; Whalen et al., 1987; Amri et al, 1996; Wu et al., 2007). Hornblende, cordierite and alkaline dark minerals are considered to be the most important and effective markers for the identification of I-type, S-type, and A-type granite, respectively (Miller, 1985). Fractionated granite is commonly characterized by the presence of the following minerals: berylite, tantalum-niobium ores, elbaite, lepidolite, or lithium muscovite (Zhu et al., 2002; Chudík et al.,
2008; Merino et al., 2013; Wu et al., 2017).

Not only do the LP exhibit low concentrations of Zr (143 to 253 ppm), Ce (83.8 to 201.3 ppm), Zr + Nb + Ce + Y (258.2 to 507.68 ppm), and $FeO^T/MgO$ ratios (2.25 to 3.44), but the XP also display low levels of Zr (85.3 to 187.2 ppm), Ce (26.7 to 90.5 ppm), Zr + Nb + Ce + Y (201.99 to 345.07 ppm), and $FeO^T/MgO$ ratios (3.88 to 8.18). Both are distinct from typical A-type granites (Zr>250 ppm, Zr + Nb + Ce + Y>350 ppm, Ce>100 ppm, $FeO^T/MgO$>16; Whalen et al., 1987).
Furthermore, the zircon saturation temperatures for the LP ($T_{Zm}$: 770 to 825 °C) and XP ($T_{Zm}$: 729 to 807 °C) are notably lower than those of typical A-type granites (> 850 °C, Watson and Harrison, 1983; Eby, 1990, 1992; Siégel et al., 2018). In addition, all compositions of the XP fall in the region of FG, while the LP plotted into the OGT in the discrimination diagrams (Fig. 8a-b). Consequently, the LP and XP may not be classified as A-type granites. The LP exhibits lower Rb contents (159 to 267 ppm, mean 201 ppm) compared to highly fractionated granites (Rb > 270 ppm, King et al., 1997). In
contrast, the XP displays high Rb contents (189 to 477 ppm, mean 307 ppm), indicating that the XP is a highly fractionated granite (Fig. 8c), whereas the LP is not. The 10000Ga/Al ratio is elevated in late-crystallizing granite when I- and S-type granites undergo highly fractional crystallization of plagioclase (Dahlquist et al., 2014). Meanwhile, samples will be positioned within the A-type granite field in the 10000Ga/Al discriminant diagrams (Whalen et al., 1987). The LP exhibits low 10000Ga/Al ratios (2.14 to 2.78), while the XP displays high 10000Ga/Al ratios (2.46 to 3.29). It is consistent with the
observation that the samples are categorized as I & S-type granite and A-type granite, respectively (Fig. 8d-e). The LP is characterized by a low A/CNK ratio (0.95 to 1.05, less than 1.1), low $P_2O_5$ content (0.07 to 0.14 wt%, less than 0.20 wt%), and high $Na_2O$ content (3.27 to 4.22 wt%, greater than 3.26 wt%). Similarly, the XP exhibits low $P_2O_5$ contents (less than 0.20 wt%) and high $Na_2O$ contents (greater than 3.26 wt%), and these characteristics differ from those of S-type granites (Chappell and White, 1974; Chappell, 1999). Moreover, all samples are positioned in the I-type granite field in the Zr versus
TiO2 discrimination diagrams (Fig. 8f). The presence of potential I-type granite indicators, such as hornblende (Fig. 3d and g, Miller, 1985), suggests that the LP should be classified as I-type granite and the XP as highly fractionated I-type granite.



## 6.2 Petrogenesis of the late Silurian intrusions

It is generally believed that there are three main origins of granite: (1) fractional crystallization of primary basaltic magma
(Cawthorn and Brown, 1976; Wyborn et al., 1987; Turner et al., 1992; Mushkin et al., 2003), (2) partial melting and
fractional crystallization of crustal material (Barbarin, 1988; Skjerlie and Johnston, 1992; Turpin et al., 1990; Patiño Douce,
1997; Chappell et al., 2012), (3) The mixing of mantle- and crustal-derived materials (Petford and Atherton, 1996;Chappell,
1999; Harris et al., 1999; Yang et al., 2006).

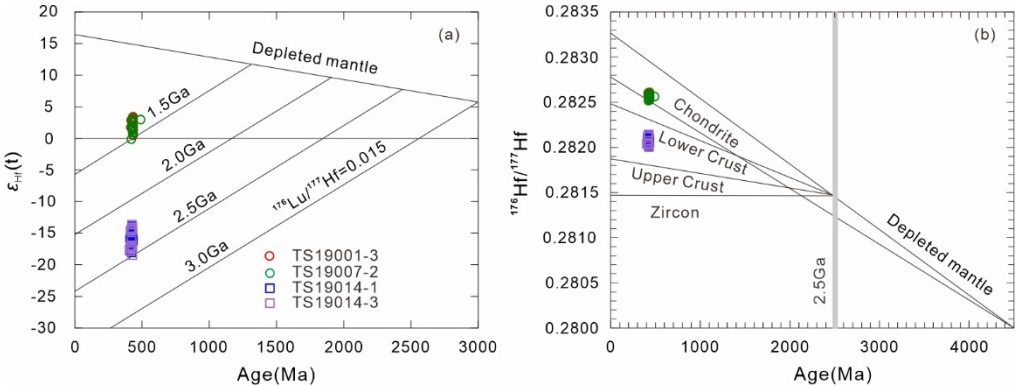


**Figure 9: Age versus $\varepsilon_{Hf}(t)$ diagram (a) and Age versus $^{176}Hf/^{177}Hf$ diagram (b) of the LP and XP in the western section of NQO.**

The LP and XP consist of K-feldspar, plagioclase, quartz, biotite and a few accessory minerals. They differ in the
following details: (1) the LP has higher $Al_2O_3$ (12.15 to 15.26 wt%), $K_2O$ (4.02 to 5.76 wt%) and $Na_2O/K_2O$ values (0.68 to
1.00) than the XP (11.27 to 14.14 wt%, 3.57 to 5.34 wt% and 0.71 to 0.91, respectively); (2) the LP has higher Sr (276 to
575 ppm) and Ba (688 to 1877 ppm) contents and Sr/Y ratios (15.67 to 53.85), and lower Y contents (8.12 to 19.49 ppm) and
Rb/Sr ratios (0.35 to 0.85) than the XP (Sr = 39 to 74 ppm; Ba = 159 to 271 ppm; Sr/Y = 0.97 to 2.62; Y = 20.43 to 42.41
ppm; Rb/Sr = 2.57 to 11.64); (3) the LP has higher La contents (44.80 to 108.25 ppm), lower Yb contents (1.89 to 19.35
ppm), and more fractional REEs ((La/Yb)$_N$ = 27.5 to 51.5; LREE/HREE = 18.9 to 30.0) than the XP (La = 12.17 to 44.17
315   ppm; Yb = 2.53 to 5.99 ppm; (La/Yb)$_N$ = 1.5 to 9.0; LREE/HREE = 2.3 to 8.9), and the Eu anomalies are slightly negative
(Eu/Eu* = 0.54 to 1.03) and significant negative (Eu/Eu* = 0.18 to 0.33), respectively; and (4) LP has positive $\varepsilon_{Hf}(t)$ values
of -0.1 to +3.4 and XP has negative $\varepsilon_{Hf}(t)$ values of -18.5 to -13.6. These differences indicate that the LP and XP formed by
different process.





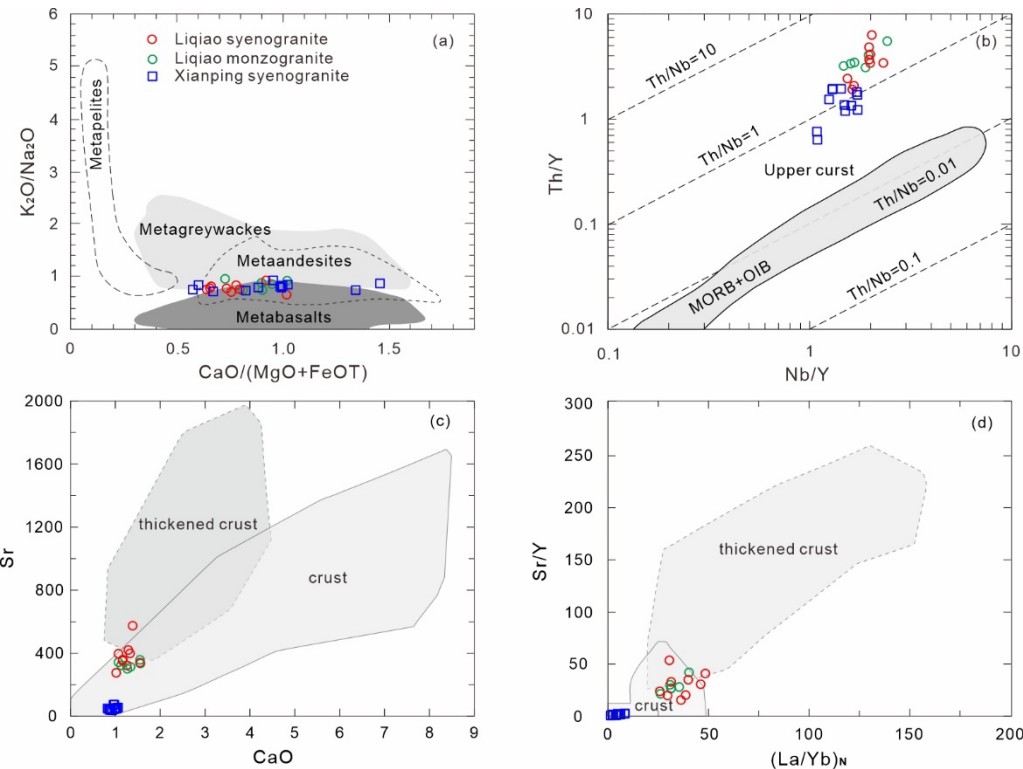

**Figure 10: The source region diagrams of the LP and XP in the western section of NQO: (a) molar K₂O/Na₂O versus molar CaO/(MgO+FeO$^T$) (Altherr and Siebel 2002); (b) Nb/Y versus Th/Y (Boztuğ et al. 2007); (c) CaO versus Sr (He et al. 2011); (d) (La/Yb) ₙ versus Sr/Y (He et al. 2011).**

### 6.2.1 Liqiao pluton

On the one hand, LP have low Na₂O/K₂O (0.68 to 1.00) and middle A/CNK (0.95 to 1.05) ratios, indicating that it was formed by biotite dehydration melting (Patiño Douce and Beard, 1995). In general, a positive value of $\varepsilon_{Hf}$ (t) can be interpreted as the source rock being from the juvenile crust or depleted mantle, whereas a negative value of $\varepsilon_{Hf}$ (t) indicates that the source rock is ancient crustal components (Taylor and McLennan, 1985; Wu et al., 2007). The $\varepsilon_{Hf}$ (t) values of the LP range from -0.1 to +3.4 (Fig. 9), which indicating that the source rock is juvenile crustal components. On the other hand, LP has high Sr (276 to 575) and La (44.8 to 108.3) contents, low Y (8.12 to 19.49) and Yb (0.88 to 1.89) contents indicated that the LP might be partial melting of thickened crust. The negative Eu anomalies (Eu*/Eu = 0.54 to 1.03) indicate that plagioclase is the predominant residue phase in the melting process (Patiño Douce and Beard, 1995). The Nb/Ta ratios (mean 14.28) and Zr/Hf ratios (mean 37.42) of the LP were close to the continental crust (Nb/Ta = 13.4, Zr/Hf = 35.7), significantly lower than melted magmas from the mantle source (Nb/Ta = 17.5) (McDonough and Sun, 1995; Rudnick, 1995; Weyer et al., 2003). The Rb/Sr ratios of the crust source magmas is generally greater than 0.5, while the Ti/Zr ratios were usually less than




20 and the Cr contents of the primitive basaltic magma were 500 to 600 μg/g (Wilson, 1989). LP has low Cr content (2.89 to 7.27 ＜500 μg/g), Ti/Zr ratios (6.12 to 7.85 ＜20) and high Rb/Sr ratios (0.35 to 0.85 ＞0.5), which indicated the LP were fundamentally different from mantle source magma and similar to crustal source magma. Also, In the LP, no mafic
intrusions have been found, and it has high $SiO_2$ contents (71.09 to 76.45 wt%), which indicated that the fractional crystallization of mantle source magma model is impossible (Defant and Drummond, 1990).

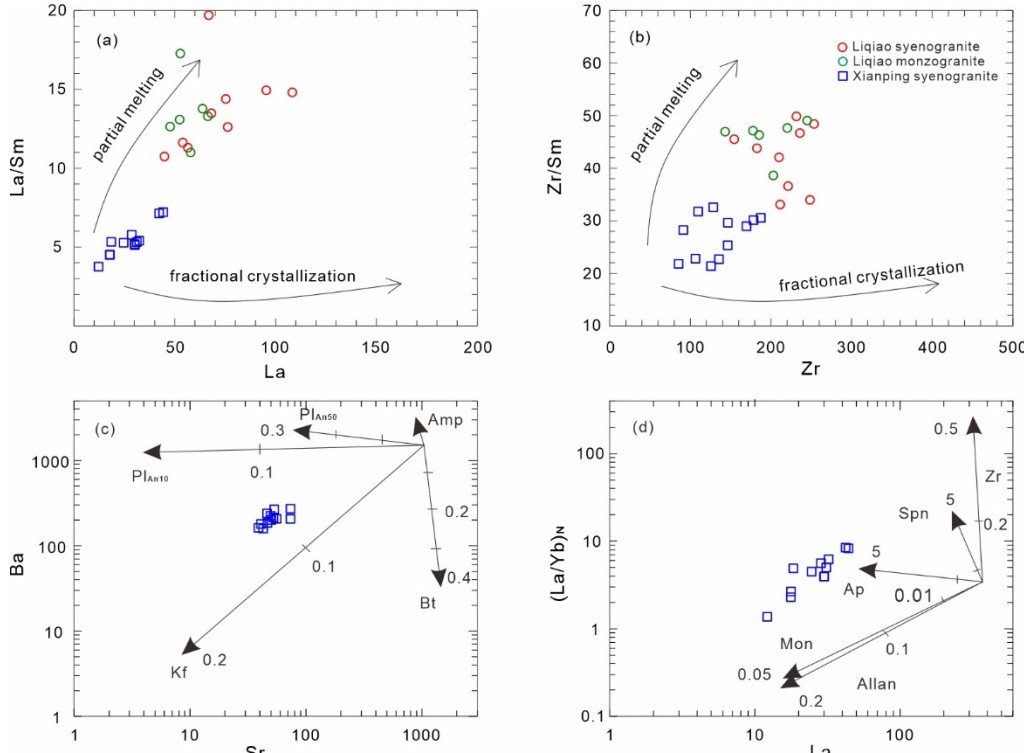

**Figure 11: The evolution and fractional diagrams of the LP and XP in the western section of NQO: (a) La versus La/Sm (b) Zr**
**versus Zr/Sm (c) Sr versus Ba (Rudnick 1995) (d) La versus (La/Yb) $_N$ (Rudnick 1995). Pl: Plagioclase; Kfs: K-feldspar; Amp: Amphibole; Bt: biotite; Zr: Zircon; Spn-Sphene; Ap: Apatite; Mon: Monazite; Allan-Allanite.**

As shown in the $K_2O/Na_2O$ versus $CaO/(MgO+FeO^T)$ diagram (Fig. 10a), the samples plotted in the field of metaandesites and metabasalts. In the Nb/Y versus Th/Y diagram (Fig. 10b), they plotted in middle-lower field. In the CaO versus Sr and (La/Yb) $_N$ versus Sr/Y diagrams (Fig. 10c-d), they plotted in crust field and there is a clear tendency to evolve
towards to the thickened crust. The positive correlations of La and La/Sm, Zr and Zr/Sm suggest that the LP exhibits partial melting trend (Fig. 11a-b). The $Mg^\#$ value can serve as a basis for determining whether mantle-derived material has been added to crustal source magma (Frost, 2001). Melt produced by crustal melting shows $Mg^\# <40$, and $Mg^\#$ does not change with the degree of partial melting, and it only displays $Mg^\# >40$ when mantle-derived material is added (Frey et al., 1978; Rapp and Watson, 1995; Hawkes and Kemp, 2006). The $Mg^\#$ values of LP range from 37.86 to 48.25 (mean 41.81>40),



indicating the involvement of mantle-derived material in the magmatic source area. In summary, we propose that the LP was formed by partial melting of juvenile felsic crust with the involvement of mantle-derived material.

### 6.2.2 Xianping pluton

There are two main sources of highly fractionated I-type granite: (1) Formation of highly fractionated I-type granite as a result of partial melting of crustal material due to magmatic underplating of the fractionated mantle source (Wu et al., 2003;

Wang et al., 2014). (2) The fractionated basic magma from mantle source underplated the lower crust and mixed with crust-derived felsic magma. They formed shallow-source hybrid magma chambers and undergo fractional crystallization to form highly fractionated I-type granite in later stage (Qiu et al., 2008). The high $SiO_2$ contents (73.92 to 78.28 wt%) of the XP suggests that it was not derived from mantle-derived magmas (Defant and Drummond, 1990). The XP showed evolved zircon Hf isotopic compositions ($\varepsilon_{Hf}(t)$ = -18.5 to -13.6), with corresponding two-stage model ages of 2259 to 2560 Ma (Fig.

9), these features clearly differ from the LP, suggesting a matured continental source region (Taylor and McLennan, 1985; Wu et al., 2007). It has higher $K_2O$ contents (3.57 to 5.34 wt%), low Nb/Ta ratios (mean 11.61) and Zr/Hf ratios (mean 27.93), similar to crustal source magmas (Nb/Ta=13.4, Zr/Hf=35.7), different from mantle source magmas (Nb/Ta= 17.5) (McDonough and Sun, 1995; Rudnick, 1995; Weyer et al., 2003). It also has low Cr content (1.42 to 5.36 <500 μg/g), Ti/Zr ratio (2.63 to 5.73 <20) and high Rb/Sr ratio (2.57 to 11.64 >0.5), suggesting that the XP was crustal source origin (Wilson,

1989). As shown in the $K_2O/Na_2O$ versus $CaO/(MgO+FeO^T)$ diagram (Fig. 10a), the samples plotted in the field of metaandesites and metabasalts. In the Nb/Y versus Th/Y diagram (Fig. 10b), they plotted in middle-upper crust field. In the CaO versus Sr and $(La/Yb)_N$ versus Sr/Y diagrams (Fig. 10c-d), they plotted in crust field. The Mg# values of XP range from 20.40 to 35.11 (<40), indicating the absence of mantle-derived material in the magmatic source area (Frey et al., 1978; Rapp and Watson, 1995; Hawkes and Kemp, 2006). Characterization of partial melting in La versus La/Sm and Zr versus

Zr/Sm diagrams (Fig. 11a-b). The XP was characterized by a high differentiation index (91.69-94.47, average 93.22) as well as significant depleting in HFSEs (i.e., Nb, Ta, P, Ti), suggesting that its source magma underwent significant fractional crystallization. Negative anomalies of Eu (Eu*/Eu = 0.18 to 0.33) and depleted in Sr and Ba in the samples were also indicative the fractional crystallization of plagioclase and K-feldspar (Rudnick, 1995; Patiño Douce and Beard, 1995). The Sr versus Ba diagram (Fig. 11c) reflects fractional crystallization of K-feldspar and plagioclase with a predominance of K-

feldspar, which is consistent with the occurrence of K-feldspar and plagioclase phenocrysts in the samples. The depleted in P and Ti, Nb, and Ta were due to fractional crystallization of apatite and Ti-rich minerals, respectively, and as shown in the La versus $(La/Yb)_N$ diagram (Fig. 11d), also reflected fractional crystallization of apatite and sphene. In summary, XP may be the formed by partial melting and highly fractional crystallization of the ancient felsic crust.



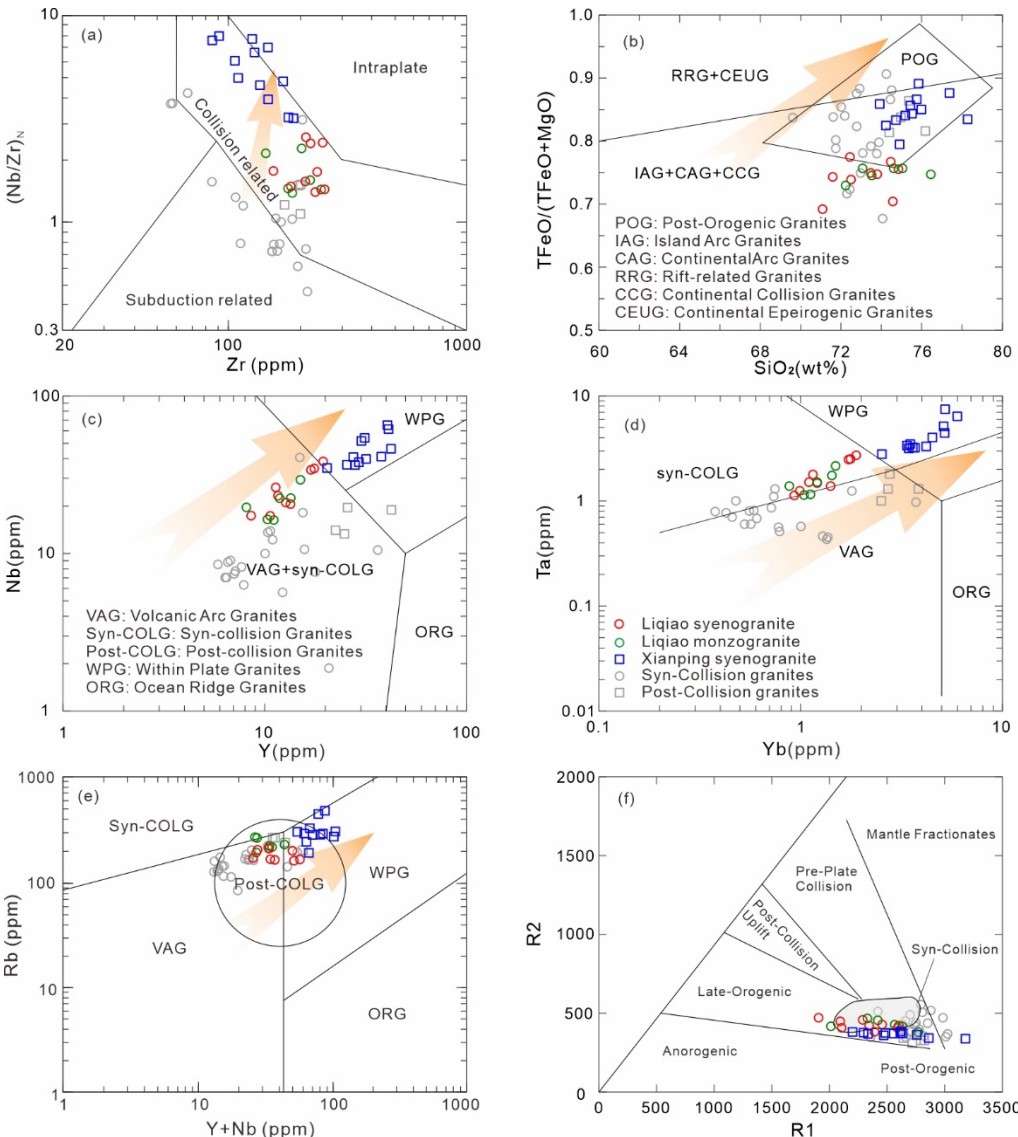

**Figure 12: The tectonic setting diagrams of the LP and XP in the western section of NQO: (a) Zr versus (Nb/Zr) $_N$ (Batchelor and Bowden 1985); (b) SiO₂ versus FeO$^T$/(FeO$^T$+MgO) (Maniar and Piccoli 1989); (c) Y versus Nb (Pearce et al. 1984); (d) Yb versus Ta (Pearce et al. 1984); (e) (Y+Nb) versus Rb (Pearce et al. 1984); (f) R1 versus R2 (Batchelor and Bowden 1985). Syn-Collision plutons from the western section of NQO after Wang et al. 2008; Ren et al. 2021; Qin et al. 2022; Xin and Huang 2023. Post-Collision pluton from the western section of NQO after Ren et al. 2021.**

## 6.3 Tectonic background

According to the variations in chronology and geochemical features in the western section of the North Qinling Orogen, Ren et al. (2021) proposed that three magmatic pulses indicate a tectonic switch from slab rollback (ca. 441–434 Ma), slab break-off (ca. 430–423 Ma), to post-collision (ca. 415–409 Ma) from 441 to 409 Ma. In contrast, Qin et al. (2022) identified three main Paleozoic magmatic pulses from ~500 to 410 Ma, which correspond to oceanic subduction (ca. 500 to 450 Ma), slab



rollback & break-off (ca. 440 to 430 Ma), and extension in a post-collision setting (ca. 420 to 410 Ma), respectively. The zircon U-Pb ages of the LP was 429 Ma, while the XP was 421 Ma, both representing the period of collision orogeny (Pei et al., 2009; Wang, 2013; Liu, 2013; Xie et al., 2020). They are consistent with the second and third magma pulses, respectively, that have been studied by previous authors (Ren et al., 2021; Qin et al., 2022). In the Zr versus (Nb/Zr) $_N$ diagram (Fig. 12a), LP and XP plotted into the region of the collision related environment, indicating that they were formed

in the collision-related setting, and the XP should be later than LP. In the $SiO_2$ versus $FeO^T/(FeO^T+MgO)$ diagram (Fig. 12b), most samples from the LP plotted into the IAG-CAG-CCG region, while the XP fell into the POG region. In the Y versus Nb and Yb versus Ta diagrams (Fig. 12c-d), the LP mostly fell into the syn-collision field. In the (Y+Nb) versus Rb diagram (Fig. 12e), the XP mainly fell into the post-collision field. The LP shows syn-collision-related features, while the XP shows late-orogenic-related features (Fig. 12f). Combining with previous research on syn-collision and post-collision granite in the

western section of the NQO (Wang et al., 2008; Ren et al., 2021; Qin et al., 2022; Xin and Huang, 2023), chondrite-normalized REE patterns and primitive mantle normalized trace element spider diagrams of the LP (Fig. 7a-b) was similar to the syn-collision granite, while the XP (Fig. 7c-d) was similar to the post-collisional granite, and there was a very clear linear evolutionary relationship from the LP to the XP, namely the trend from syn-collision to post-collision (Fig. 12). It can be inferred that the LP formed in syn-collision stage (Fig. 13a), as well as the XP formed in a post-collision stage (Fig. 13b).

Both were products of the collision-orogeny between the northern margin of the YB and the NQO.

### 6.4 Tectonic significance

The Proto-Tethys Ocean was a giant ocean developed during Neoproterozoic to Early Paleozoic, which was located between the northern Laurasia and the southern Gondwana, and it generated by the breakup of the Rodinia Supercontinent (Mattern and Schneider, 2000; Stampfli and Borel, 2002; Raumer and Stampfli, 2008; Li et al., 2016b, 2018b; Yang et al., 2018; Wu

et al., 2020). The northern boundary of the Qinling-Qilian Orogen in the CCOB is regarded as the northern boundary of the Early Paleozoic Proto-Tethys Ocean closure (Li et al., 2016b, 2017). The NQO preserves abundant Early Paleozoic records, revealing the tectonic evolution of the northern Proto-Tethys Ocean on the periphery of the northern margin of eastern Gondwana (Dong et al., 2021; Mark et al., 2023). The WSS is considered to represent an oceanic suture zone formed by closure of the WSO, which formed the northern part of the Proto-Tethys Ocean along the northern margin of the eastern

Gondwana during the Early Paleozoic (Zhang et al., 2001; Pei et al., 2009; Chen et al., 2024). The WSS is the most important tectonic zone separating the NCB from the YB (Zhang et al., 2001; Yang et al., 2002; Xu et al., 2006; Pei et al., 2009). The mate-basic volcanic rocks within Guanzizhen and Wushan ophiolite exhibited features of N-MORB and E-MORB, respectivelt, and yield zircon U-Pb ages of 534 to 489 Ma, which were both represented the remnants of the WSO lithosphere (Hou et al., 2006; Li et al., 2007; Li, 2008; Pei et al., 2004, 2007a, 2009; Dong et al., 2008, 2011a). The

northward subduction of the WSO in the western section of the NQO in the Ediacaran-Ordovician resulted in the collision of the YB and the NQO in the southern part of the NCB, which reflected the relationship between the Caledonian orogeny in



the NQO and the tectonic evolution of the Proto-Tethys Ocean (Pei et al., 2005, 2009). In combination with the results of previous research, the tectonic evolution of the western section of the NQO can be divided into three stages as follows.

**Figure 13: Schematics geodynamic and petrogenetic model illustrating the evolution of the western section of the NQO at ca. 429 Ma and 421Ma, and the generation of the LP (a) and XP (b) during Late Silurian. MORB—mid-ocean-ridge basalt.**

Stage 1 from 472 Ma to 438Ma. Northward subduction of the WSO.

With the northward subduction of the NQO, the Liziyuan Group represented by the metamorphic-volcanic-sedimentary rock system of the island arc-forearc basin during 472 to 451 Ma, and the Caotangou Group represented by the island-arc-type volcano-sedimentary rock system was formed in the fore-arc basin, which located between the island arc and the



subduction zone at the southern margin of the NQO (Pei et al., 2006; He et al., 2007; Yan et al., 2007; Yang et al., 2018a). The Liushuigou gabbro, located north of the Guanzizhen ophiolite, has a zircon U-Pb age of 471 Ma, which was formed in
island-arc tectonic setting (Pei et al., 2005, 2009; Yang et al., 2006). Additionally, the Baihua intermediate-basic igneous complex (449.7 Ma) in the western section of the NQO, the Hualingou meta-gabbro (440 Ma) and the Yuanyangzhen meta-gabbroic amphibolite (456 Ma) in the Wushan area were formed in an island-arc tectonic setting (Pei et al., 2007b; Li, 2008). The intermediate-basic volcanic (456.4 Ma, e.g. the Zhangjiazhuang Formation) and the acidic tuffs (457.4 Ma, e.g. the Longwanggou Formation) showed arc magmatism characterize, which recorded the subduction of WSO in the Early
Paleozoic (Wang et al., 2007; Chen et al., 2019; Zhu et al., 2008; Xu et al., 2014). There were Honghuapu tonalite (450.5 Ma), Tangzang quartz diorite (454.7 Ma), and the Sanchahe quartz diorite (459 Ma) in the Fengxian area (Wang et al., 2006, Chen et al., 2008, Qing et al., 2022), as well as the Yangjiazhuang and Honghuapu K-rich, high-Ba-Sr granitoids (439-438 Ma, Ren et al., 2018), were formed in subduction-related setting. In this stage, the slab roll-back of the WSO initiated asthenosphere decompression melting in the mantle wedge and generated corresponding magmatic activities. Subsequently,
asthenosphere convection induced partial melting of enriched mantle that had interacted with slab-derived fluids and produced basaltic magma that evolved into the Early Ordovician-Early Silurian gabbro, diorite and quartz diorite by fractional crystallization. Slab roll-back and its consequences strongly increased the geothermal gradient of the lower crust, which formed the early Silurian granites. These arc-related and subduction-related magmatic rocks define the age for the northward subduction of the WSO from 472 to 438 Ma.

Stage 2 from 438 Ma to 423Ma. Initial continental collision of the YB with the NQO.

Previous studies have shown that slab break-off usually occurs very soon after continental collision (Duretz et al., 2011; van de Zedde and Wortel, 2001). Continuous subduction ultimately gave rise to the consumption of the WSO oceanic crust and final collision between the YB and the NQO in the Early Silurian which resulted in the significantly thickened the crust, and there was no typical sedimentary state exposed in the area during this stage, but the intrusion activities of collisional-
related magmas was violent with extensive development of Caledonian pluton (Pei et al., 2009). Due to buoyancy differences, slab break-off might have taken place, followed by subsequent asthenosphere upwelling, as well as the asthenosphere mantle-derived magma supplied both heat and mantle material, which underplated the lower part of the accretionary prism and triggered partial melting of juvenile crustal material accompanied by crust-mantle interaction (Davies and von Blanckenburg, 1995). The Dangchuan granite was formed by the partial melting of the thickened lower
crust, with a U-Pb age from 438 to 432 Ma (Wang et al., 2008; Qing et al., 2022; Xin and Huang, 2022), as well as the Xiongshangou syn-collision granite (named LP of this paper) formed at 438 Ma (Wang, 2013), and the Jiguanya syn-collision granite at 435 Ma (Xu et al., 2018). The Tangzang Na-rich high-Ba-Sr granite, Zhangjiazhuang two-mica granite and muscovite granite formed during the slab break-off from 430 to 423 Ma (Ren et al., 2018, 2021). The formation of the LP (429 Ma) studied in this paper was attributed to partial melting of lower crust dominated by accreted WSO oceanic crust
and sediments, accompanied by local magma mixing with the mantle-derived magma. The presence of these intrusive rocks suggested that the WSO closed before 438 Ma, and that the northern margin of the YB and the NQO were underwent





continent-continent collision from 438 to 423 Ma. During the Late Silurian (433 to 424 Ma), the western section of the NQO experienced granulite-facies metamorphism and anatexis (Mao et al., 2017), which may have occurred in a thickened lower crustal setting during the early stage of collision orogeny (Yu et al., 2013; Peng et al., 2021). This reflects the existence of an Early Paleozoic crustal thickening in the western section of the NQO, which comprehensively indicates that the northern margin of the YB and the NQO were in a docking and collision stage during the period of 438 to 423 Ma.

Stage 3 from 421 Ma to 409 Ma. Post-collision extension phase.

Given that slab break-off occurred before 423 Ma in the western section of the NQO (Ren et al., 2021), delamination of thickened crust might have caused extension during the Late Silurian-Early Devonian. Continued continental compression from 438 to 423 Ma would cause shortening and thickening of the continental crust and underlying lithospheric mantle to produce a high-density lithospheric root (eclogitic root) that protruded into the asthenosphere (Houseman and Molnar, 1997). The delamination was characterized by the sinking of eclogitic root and subsequent upwelling of extensive hot asthenosphere, accompanied by crustal extension (Deng et al., 2015; Ma et al., 2015). Huoyanshan granite was formed by high-temperature melting of the heterogeneous continental crust under extensional environment, with a zircon U-Pb age of 415 Ma (Qing et al., 2022). Leijayuan quartz diorite (415 Ma), Beixinggou black mica granite (412 Ma) and Yanwan dolomite granite (409 to 414 Ma) were formed in a post-collision tectonic setting (Wang et al., 2009; Ren et al., 2021). They record the tectonic evolution of the post-collision phase of the NQO. The XP (421 Ma) in this paper was product of the post-collision orogeny stage of extension between the northern margin of the YB and the NQO. The upwelling of hot asthenosphere underplated the ancient crust and supplied heat, not only caused the crust to thin and decompress, but also resulted in widespread partial melting of the ancient intermediate-basic lower crust. Later, under the control of mineral separation crystallization processes such as K-feldspar, plagioclase, and apatite, it underwent highly fractional crystallization to form the Xianping highly fractionated I-type granite. Reflecting the information on the differentiated evolution of crustal materials during the Late Caledonian post-collision in the western section of the NQO, it provides further evidence to reveal the subduction-collision timeframe of the WSO in the NQO during the Early Paleozoic.

In the global context, the Proto-Tethys Ocean was located between the North American Laurentia-Tarim-North China and Gondwana continent, formed by the breakup of the Rodinia supercontinent (Stampfli and Borel, 2002; Wu et al., 2020; Dong et al., 2022). The Rodinia supercontinent underwent a breakup during the Neoproterozoic, and the Laurentia and Baltic continent separated to form the Iapetus Oceans in the Late Proterozoic, which face west towards the continent of West Gondwana in the Southern Hemisphere (van Staal et al., 2012). In the Late Ordovician, the Avalonia Terrane drifted northwards to form the Rheic Ocean in the active continental margin of the western Gondwana, which continued to expand and shrink the Iapetus Ocean, while the collision of the Avalonia Terrane with the Baltic continent caused the disappearance of the Tornquist Ocean and the formation of the Thor Suture (Torsvik and Rehnstrom, 2003). Subsequently, the collision of the Avalonia Terrane-Baltic continent with the Laurentia continent formed the Appalachian-Caledonian orogen in southern North America-Britain-Norway, which resulted in the disappearance of the Iapetus Ocean between the Laurentia and Baltic continents, it means the closure of the Proto-Tethys Ocean (McKerrow et al., 2000; Dewey et al., 2015; Torsvik, 2019). In



the northern part of the British Caledonian Belt, between the Baltic and Greenland continent, the famous Norwegian HP-UHP metamorphic rocks were formed due to the subduction of the Baltic under the Laurentia continent, which indicated that the Proto-Tethys Ocean had disappeared at ca. 435 Ma (Wu et al., 2020).

The CCOB is one of the most developed locations in the Proto-Tethys tectonic domain. Through long-term research on this area, it has been found that the ages of ophiolites in the North Altyn-North Qilian-Kuanping Suture, South Altyn-North Qaidam-Shandan Suture, and Kudi-Kunzhong Suture were mainly concentrated between 520~490 Ma (Xiao et al., 2003; Zhang et al., 2004, 2015; Song et al., 2019). The Kangxiwar-A'nyemaqen-Mianlue Suture zone developed a large number of Early Paleozoic ophiolites of 530~480Ma and Late Paleozoic ophiolites. A small amount of Proto-Tethys ophiolites also developed in the Longmuco-Shuanghu-Changning-Menglian belt (Pan et al., 2020; Wang et al., 2021). These ophiolites

reflected the formation of the Proto-Tethys Ocean in China, while the formation of the European Iapetus Ocean is basically consistent with or slightly earlier than this (Xu et al., 2010a, 2010b; Song et al., 2013, 2019; Wu et al., 2020; Li et al., 2022). The NQO marks the northernmost boundary of the Proto-Tethys Ocean in the eastern part of the CCOB (Li et al., 2016a), as well as the WSO is located in the northern part of the Proto-Tethys Ocean (Zhang et al., 2001). The closure of the WSO resulted in the collision of the northern margin of the YB with the NQO in the Silurian, which ultimately merged with the

eastern part of the Gondwana continent, and were closely related to the evolution of the Proto-Tethys Ocean (Zhao et al., 2018; Li et al., 2018b).

## 7 Conclusions

(1) LP and XP both were high-K calc-alkaline metaluminous-weakly peraluminous granite series, which showed high $K_2O$, low $Na_2O$, $Fe_2O_3^T$, and exhibited distinct enrichment in LFSE (e.g. Rb, K, and Pb), and depleted in HFSE (e.g. Nb, Ta,

Ti, Zr, and Ce). The LP has higher REE contents (180.35 to 412.32 µg/g), more fractional REEs ($(La/Yb)_N$ = 27.5 to 51.5; LREE/HREE = 18.9 to 30.0) than the XP (REE = 81.26 to 205.05 µg/g, $(La/Yb)_N$ = 1.5 to 9.0, LREE/HREE = 2.3 to 8.9), and the Eu anomalies are slightly negative (Eu/Eu* = 0.54 to 1.03) and significant negative (Eu/Eu* = 0.18 to 0.33), respectively;

(2) The zircon U-Pb age of the LP and XP were 429 Ma and 421Ma, respectively. The LP was I-type granite with

positive $\varepsilon_{Hf}(t)$ values of -0.1 to +3.4, which was formed by partial melting of the intermediate-basic juvenile lower crust, accompanied by magma mixing with the mantle material. The formation of the Xianping highly fractionated I-type granite ($\varepsilon_{Hf}(t)$ = -18.5 to -13.6) was attributed to partial melting of mature lower crust with highly fractional crystallization in extensional setting.

(3) From the Liqiao I-type granite to Xianping highly fractionated I-type granite in the western section of the NQO,

reflecting the transformation of the thickening of the crust and the magma mixing of crust-mantle materials in syn-collision stage between the YB and NQO to post-collision stage with fractional crystallization of crustal material. They were closely related to the evolution of the WSO, northern of the Proto-Tethys Ocean.



## Data availability

The data presented in the supplement.

## Author contribution

Hao Lin: Conceptualization, Formal analysis, Investigation, Methodology, Visualization, Writing – original draft. Zuochen Li: Conceptualization, Formal analysis, Investigation, Methodology, Visualization, Funding acquisition. Xianzhi Pei: Investigation, Methodology, Funding acquisition. Meng Wang: Investigation. Shaowei Zhao: Formal analysis, Methodology. Hai Zhou: Formal analysis, Methodology. Feng Gao: Investigation. Mao Wang: Writing – review & editing, Methodology. Li Qin: Investigation, Methodology.

## Acknowledgements

We thanks are extended to the chief editor and the two anonymous reviewers for their constructive reviews which have greatly improved our manuscript. We wish to acknowledge Yifeng Wang and others for their help during the fieldwork and sample preparations. We also extremely thankful to Dr. Yinchuan Wang and Dr. Xiao Wang provided valuable comments on the language of this paper.

## Financial support

This work was supported by the National Natural Science Foundation of China [Grant number 41872235, 42172236, 41872234]; Fundamental Research Funds for the Central Universities [Grant number 300102270202, 300103183081, 300104282717]; and Youth Innovation Team of Shaanxi Universities.

## Competing interest

The authors declare that they have no conflict of interest.

## Disclaimer

Publisher's note: Copernicus Publications remains neutral with regard to jurisdictional claims made in the text, published maps, institutional affiliations, or any other geographical representation in this paper. While Copernicus Publications makes every effort to include appropriate place names, the final responsibility lies with the authors.



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
