# Peer review of "Silurian syn- and post-collision granitic magmatism in the western section of the North Qinling Orogen: Implications for collisional orogenic processes"

_EGUsphere, 2024_

## Author Comment (AC1)

I am deeply grateful to Mark Allen for your thoughtful comments on this article, as well as for the invaluable advice you have shared. Your feedback has been extremely constructive and instrumental in enhancing our work. We have addressed all the points that we found insightful, which will undoubtedly strengthen our paper. Our detailed responses to each of the reviewer's comments and suggestions are shown below.

Review of Silurian syn- and post-collision granitic magmatism in the western section of the North Qinling Orogen: Implications for collisional orogenic processes

By Lin et al.

This paper analyses samples from the Liqiao and Xianping plutons in the Qinling orogenic belt. The geochronology and geochemistry studies are good, and make a case for the plutons having similar chemistry - I-type granite, with ages of 429 Ma and 421 Ma respectively. But, at present there is a bit of a "so what?" feel to the paper – the results are not a surprise, and the tectonic interpretation has problems which I'll cover later in the review.

The regional review is okay, but any and all attempts to summarise the geology of the Qinling are open to criticism simply because we don't yet have a unified understanding of the range and its evolution.

Have another go at drafting figures 1a and 1b – the boundaries don't quite match between the two parts. Also, be clearer about the correlations that are intended. There is a broad match of colours between the Qilian/Qaidam/Kunlun regions to the west and the units of the Qinling, but it is not clear what is meant to be correlated and what is not. The Qinling map can be improved to add more of the important tectonic units, such as the Erlangping Unit and the Shangdan Suture Zone – both of these are wide enough to be marked on Figure 1, and they are important in the regional tectonic evolution.

Response: We have modified figures 1. The study area of this paper is the western section of the North Qinling Orogeny. The Kuanping Unit, Erlangping Unit and Shangdan Suture zone have been added to figure 1a.

[Figure]

**Figure 1: (a) Simplified geological map of the division of tectonic units in Qinling Orogen (Mao et al. 2017; Dong et al. 2022b) (b) Simplified geological map of the conjunction zone between the North Qinling Orogeny and North Qilian Orogeny showing the distribution of Early Paleozoic plutons (Pei et al. 2004; Xu et al. 2012)**

There are few comments or criticisms to make on the geochronology and geochemistry parts of the paper. They are done well in my view. It would be good to have more description of the field relations and contacts. Are the samples foliated in any way? This is not mentioned, so presumably not, but there are hints of grain alignments in figures 3b and 3f.

Response: The original structures observed in Xianping pluton are believed to have been formed during magmatic crystallisation process. During the process of magmatic crystallisation, the magma at the edge of the Xianping pluton remains thermoplastic due to the influence of its melt. The upwelling magma, along with the pressure of surrounding rocks, results in the formation of local

shear, leading to mylonitization of surrounding rocks, the minerals are flattened and elongated. This causes the minerals to become flattened and elongated, and it also leads to the formation of augen structures and banded structures.

The tectonic interpretation needs more thought, but I don't expect full agreement with everything I suggest here.

First, please define what you mean by "syn-collision" and "post-collision" – this is central to the paper as the phrases appear in the title. But, it is not clear what is meant by each term. Also think about "orogeny". Collisions can last for 10s of millions of years, and orogenies for even longer as they typically include a phase of pre-colllisional, oceanic, subduction. The India-Eurasia and Arabia-Eurasia examples show us that plate convergence can last for 10s of millions of year after *initial* continental collision. Presumably all the magmatism and deformation associated with this overall convergence is still "syn-collisional". But, during this time the overriding plate can experience episodes of extension (See Tibet), tectonic escape (Anatolia) and possibly slab break-off, or delamination – all while convergence continues. So, how can we define "post-collisional"? I'll argue that the term should be kept for processes and events that take place after overall plate convergence has stopped, and should relate to a different tectonic cycle and setting. I think "post-collisional" commonly gets applied to magmatism and deformation that is very much part of the orogeny and continued convergence. See Şengör's papers for more discussion of these issues, e.g. "Plate tectonics and orogenic research after 25 years: Synopsis of a Tethyan perspective".

Response: We have defined the "syn-collision" and "post-collision" in section 6.3 from line 398 to 415. The orogeny is typically used to describe a range of geological processes, including the magmatism of island and continental arcs, oceanic subduction and closure, continents collision and subduction, slab rollback during the syn-collision, delamination and orogenic collapse during the post-collision (Song et al., 2015; Xu et al., 2021; Zhang and Hou 2015; Zheng et al., 2015, 2022; Zhu et al., 2022). The initiation of continental collision occurs when two continents converge and the oceanic crust between them is fully subducted, and the demise of this oceanic crust can be inferred from the ages of the high-pressure (HP) metamorphic rocks, arc volcanic rocks, and youngest ophiolites (Song et al., 2015). The syn-collision is characterized by a continental collision and subduction process. Despite the ocean having closed, subduction is not terminated, this is because it requires a significant amount of time for one continent to be subducted beneath another and achieve ultra-high-pressure (UHP) metamorphic conditions. During the process of continental collision and subduction, both the previously subducted oceanic crust and the continental crust undergo decompression melting as they are exhumed and the crust thickens, and accompanied by partial melting of crustal material to produce tonalites and peraluminous granites, without input of mantle materials (Chen et al., 2013; Liu et al., 2014; Song et al., 2014). Additionally, crustal melts derived from the deep subducted continental crust during exhumation could metasomatize the overlying lithospheric mantle, leading to the formation of mafic magmas (Zhao et al., 2012). The post-collisional is used to describe magmatism that occurs after the major collisional event, which typically marks the end of orogeny. It occured in the relaxation phase following the main orogeny, characterized by the large-scale horizontal movement of plate boundaries in the early phase and, in the late phase, by the delamination and collapse of orogeny, extension of the lithosphere, and crust-mantle interactions, such as the post-collisional magmas in the North Qaidam UHP metamorphic

belt (Wu et al., 2014; Wang et al., 2014).

In this paper, there are two similar plutons in terms of chemistry and timing (only 8 Myr) apart, and yet one is assigned to a "syn-collisional" setting and the other to a "post-collisional" setting. The older Liqiao pluton is shown in figure 13a and being linked to slab break-off, the younger Xianping pluton is shown in figure 13b as being linked to delamination. Note that there is not independent evidence for either of these popular tectonic processes having taken place at these times. Given that the broadly I-type chemistry of both plutons is typical of Andean-type subduction zones, it is a simpler interpretation to assume that both plutons took place in this tectonic setting, without the need for further complications. There is an obvious objection to this scenario, in that there is evidence for collision that pre-dates both plutons, e.g. UHP metamorphism of continental protoliths in other parts of the North Qinling. But, these events can be accommodated by models where collision of microcontinents along the Proto-Tethyan margin did not terminate subduction. See Allen et al (2023) and Li Sanzhong et al (2018) for alternative scenarios, that are not simply the 2D cross-sections typically adopted by studies of local parts of the Central China Orogenic Belt. There are surely other and better models waiting to be developed.

Response: We agree with your model where collision of microcontinents along the Proto-Tethyan margin did not terminate subduction. The Qinling Complex in the western section of the North Qinling Orogeny underwent late Silurian (433−424 Ma) granulite-facies metamorphism with a clockwise *P-T* path, and followed by 411−402 Ma amphibolite-facies or retrograde overprinting metamorphism, suggesting that the granulite-facies rocks may have been formed by the continent-continent collisional orogeny in the Paleozoic (Guo et al., 2022; Mao et al., 2017, 2018). In contrast, the ages of granulite-facies metamorphism in the Qinling Complex, located in the eastern part of the North Qinling Orogeny is mainly concentrated between 430 and 450 Ma (Liu et al., 2013; Xiang et al., 2014; Zhang et al., 2009, 2011). The zircon U-Pb age of the Liqiao pluton is 429 Ma, coinciding with the timeframe of magmatism during the second pulse, specifically the slab break-off (Ren et al., 2021; Qin et al., 2022). This aligns with the granulite-facies metamorphism period at 433-424 Ma (Mao et al., 2017, 2018), suggesting that the Liqiao pluton likely formed during the slab break-off phase of the syn-collisional process. Pay attention, the collision of Yangtze Block and North Qinling Orogeny along the Wushan-Shangdan Ocean margin might not terminate subduction (Li et al., 2018b, 2018c; Allen et al., 2023). The zircon U-Pb age of the Xianping pluton is 421 Ma, coinciding with the post-collisional orogenic process within the error range (420-409 Ma; Ren et al., 2021; Qin et al., 2022). It is also close to the timeframe limit of the amphibolite-facies or retrograde overprinting metamorphism from 411 to 402 Ma, suggesting that the Xianping pluton was formed during the post-collisional stage.

I don't fully understand Figure 13. If the labelled accretionary complex in the south relates to the oceanic plate subduction, what is the unlabelled continental tract immediately to its north and to the south of the remnant slab shown in 13a? How and why does this remnant slab disappear in the delamination event shown in 13b? As noted above, there seems to be no independent evidence for either slab break-off or delamination at the times and places shown. Given the simplicity of the chemistry of the I-type granites in the study, why are they not evidence for active oceanic subduction

at this time – similar to the widespread magmatism of ~430-420 Ma in many other parts of the Central China Orogenic Belt.

Response: We have modified figures 13. Both of the Liqiao pluton and Xianping pluton represent periods of collisional orogeny. We have already responded to the discussion of the Liqiao and Xianping pluton tectonic setting in the previous comment.

[Figure]

Figure 13: Schematics geodynamic and petrogenetic model illustrating the evolution of the western section of the North Qinling Orogeny at ca. 429 Ma and 421Ma, and the generation of the Liqiao pluton (a) and Xianping pluton (b) during Late Silurian.

References:

A couple of test references at the start of this section need to be deleted.

Response: We have deleted the references at the start of this section.

"Mark" et al should be "Allen" et al – here and in the main text.

Response: We have fixed this error.

Figures:

Clear and appropriate. The tectonic cartoons need a re-think, as described above.

Response: We have modified figures 13.